# Investigation of the Variability of Alkaloids in *Buxus sempervirens* L. Using Multivariate Data Analysis of LC/MS Profiles

**DOI:** 10.3390/molecules27010082

**Published:** 2021-12-23

**Authors:** Lara U. Szabó, Thomas J. Schmidt

**Affiliations:** Institute of Pharmaceutical Biology and Phytochemistry (IPBP), PharmaCampus, University of Münster, Corrensstraße 48, D-48149 Münster, Germany; lszabo@uni-muenster.de

**Keywords:** *Buxus sempervirens* L., *nor*-cycloartane alkaloids, seasonal variability, optimal harvesting time, multivariate data analysis, principal component analysis, volcano plot, mass spectrometry, fragmentation pattern

## Abstract

*Buxus sempervirens* L. is a common ornamental plant in southern and central Europe, and has been used ethopharmacologically against a wide variety of diseases due to it containing *nor*-triterpene alkaloids of the *nor*-cycloartane type. Recently, we demonstrated the interesting antiprotozoal potential of some of these compounds. To characterize the temporal variability in the alkaloid profile of two different varieties and their leaves and twigs, 30 different extracts of *B*. *sempervirens* were evaluated by Ultra High Performance Liquid Chromatography/positive Mode-Electrospray Ionization Quadrupole Time-of-Flight-Tandem Mass Spectrometry (UHPLC/+ESI-QqTOF-MS/MS). The analytical profiles were thoroughly investigated by various methods of multivariate data analysis (MVDA). A principal component analysis (PCA) model elucidates the seasonal variation in the phytochemical composition of *B*. *sempervirens* var. *arborescens* and *suffruticosa* along with differences between the varieties. Analysis of a volcano plot illustrated the differences between the two organs, the leaf and twig. Eighteen compounds were highlighted by the models as constituents of the plant characteristic for a season, variety or organ. These compounds were dereplicated based on their chromatographic and +ESI-QqTOF-MS and –MS/MS data. In addition, mass spectral fragmentation pathways for already known alkaloids as well as new natural products could be postulated for the first time. In conclusion, the MVDA models give detailed information on the temporal variability in the alkaloid profile of two different varieties and their organs (leaf vs. twig) of *B*. *sempervirens*. Thus, the results of this study allow, e.g., the identification of characteristic compounds for the different varieties, plant organs, seasons, and the optimal harvesting time for the isolation of particular *Buxus*-alkaloids of interest for subsequent studies.

## 1. Introduction

The common or European box or boxwood, *Buxus sempervirens* L. (Buxaceae), has been used as a medical plant to cure, amongst other diseases, rheumatism and malaria [1,2]. The selective in vitro activity of an organic *B*. *sempervirens* leaf extract, as well as of alkaloidal fractions and isolated alkaloids against *Plasmodium falciparum*, the causative agent of Malaria tropica, has been demonstrated by us in previous studies [3,4] Therefore, this plant can be considered as a valuable source of bioactive compounds. The purpose of this study is the investigation of differences in the alkaloid profile of two varieties (var. *arborescens* and *suffruticosa* [5] of *B*. *sempervirens,* the seasonal variability in their phytochemical composition as well as differences in alkaloid profiles between their leaves and twigs. The majority of marketed boxwood plants belong to var. *arborescens*, which has been cultivated for centuries **[6]**. This bushy evergreen shrub or small tree has pointed dark green leaves. In contrast, *Buxus sempervirens* var. *suffruticosa* exhibits broader and rounded foliage and forms small dense bushes [6].

As known from the literature, *B. sempervirens* contains characteristic aminocycloartanoid alkaloids in the leaves, twigs, flowers and roots [7]. The total alkaloid content depends on age and season [8]. The highest amounts of alkaloids were detected in first-year shoots [9].

Systematic and detailed investigations on the seasonal changes in boxwood constituents, including differences between the two mentioned varieties and the different organs of their aerial parts (leaves, twigs), to the best of our knowledge, have not been conducted previously.

To gain an indication of the characteristic compounds responsible for the major chemical differences between the two varieties, the seasonal variation in their constituents as well as the differences between plant organs, 30 extracts of *B*. *sempervirens* were prepared and examined by UHPLC/+ESI-QqTOF-MS/MS (henceforth termed “LC/MS”). Besides simple and straightforward analysis of the occurrence of certain alkaloids of interest in the various samples (Section 2.1), a multivariate data analysis (MVDA) approach was chosen in order to extract the relevant information from the large LC/MS dataset (Section 2.2 and Section 2.3). Principal component analysis (PCA) is a widely used computational method of MVDA in which an extensive number of original variables, which were determined, e.g., with an analytical procedure, and many of which contain related information, are reduced to a few so-called latent variables with unique and non-redundant information content (principal components; PCs) [10]. Since there are usually much fewer PCs than original variables necessary to encode the relevant information content of the data matrix, the dimensionality of the dataset is often greatly reduced. The set of original data (e.g., samples) can thus more easily be assessed for similarities and differences by the projection of the samples’ coordinates (scores) into a space defined by the PCs as coordinate axes (latent variable space). Thus, for instance, groupings of similar samples and the main reasons for their vicinity (meaning similarity) can be identified much more easily than by analyzing the highly complex multidimensional space defined by the original variables [11]. PCA has already been used successfully to compare five different *Buxus* varieties and to identify typical constituents [12]. In the present work, PCA is applied to highlight for the first time the seasonal variation in the phytochemical composition of two varieties of *B*. *sempervirens* (var. *arborescens* and *suffruticosa*). Additionally, another MVDA method, a volcano plot, is applied to analyze the difference between their aerial organs, i.e., leaves and twigs.

## 2. Results and Discussion

Over a period of one year (February 2019–January 2020), the shoot tips of a box hedge were harvested every month to create a seasonal profile of *B*. *sempervirens* L. In addition to the harvesting time, a distinction was made between the two varieties, var. *arborescens* and var. *suffruticosa*, and their leaves and twigs. The plant material was extracted with dichloromethane and analyzed by LC/MS to record the analytical fingerprints. Thirty extracts (12 leaf extracts from each variety for each month and 3 twig extracts from each variety for the months of February, September and December) were analyzed twice. One sample (BS01_20_B_s: January 2020, *B*. *sempervirens* var. *arborescens*) was used as a quality control (QC) which was analyzed repeatedly (after every 20 sample injections) in each measurement sequence, so that data from 63 analyses in total were acquired. The LC/MS data of all analyses were calibrated and converted into a 63 × 1200 data matrix (“bucket table”) consisting of 63 rows (analyses obtained from the 30 samples) and 1200 columns (t_R_: *m/z* variables) using the software ProfileAnalysis 2.1. The bucket table served as the basis for the following evaluations.

### 2.1. Occurrence of the Previously Isolated Buxus-Alkaloids in the Different Extracts

The bucket statistic view in ProfileAnalysis 2.1 (see Figure 1; Figure 2) shows the respective intensity of a bucket variable and the underlying compound in the 63 different MS chromatograms. Firstly, this view was used to detect the differences in content of the 25 alkaloids (**1–25**) that we isolated in a previous study and investigated for their antiprotozoal activity [3]. During this work, a mixture of *B*. *sempervirens* var. *arborescens* and var. *suffruticosa* leaves in which the variety *arborescens* clearly predominates in proportion was obtained by cutting the box hedge under study and used as starting plant material for the isolation. Therefore, it was of interest to find out whether the various constituents only appear in one particular variety or plant organ, and in which month the maximum content is present. The results of the evaluation of the bucket statistics plots are summarized in Table 1.

The data (Table 1) indicate that the studied boxwood population exhibits an increased alkaloid content in summer compared to the other months. Except for compounds **7**, **18** and **20**, all alkaloids show their maximum content in the months of July or August. The alkaloids serve the plant as a chemical defense against herbivores [12]. Therefore, it makes sense for the *Buxus* plants to adapt the production of their defensive substances to the appearance of predators, which are much more active in summer than in winter.

Furthermore, it can be concluded that the two varieties show a different composition of the alkaloid pattern. Some compounds occur only in one of the two varieties (var. *arborescens*: **1**, **3**, **4**, **10**, **23**, **25**; var. *suffruticosa*: **5–7**), while others differ significantly in content (**2**, **8**, **9**, **11–14, 17**, **19–21**, **24**) or are almost equally represented in both varieties (**15**, **16**, **18**, **22**).

The constituents were mainly detectable in the leaves, but also partly in the twigs.

#### Optimal Harvesting Time

*B*. *sempervirens* is a very popular ornamental plant in southern and central Europe that needs to be trimmed regularly. Hence, its leaves are a sustainable source of *Buxus*-alkaloids.

Through the bucket statistic view in the software ProfileAnalysis 2.1, the optimal plant starting material can be selected for a subsequent isolation, in which the target compound is present in *B*. *sempervirens* in the highest possible content. The time of harvest, the variety and the organ can be decisive for the yield of the substance to be isolated.

For instance, the optimal harvesting time for the isolation of *O*-tigloylcyclovirobuxeine-B (**1**), a *Buxus*-alkaloid with promising antiplasmodial activity (IC_50_ 1.05 µM (*Plasmodium falciparum*) vs. 19 µM (cytotoxicity); selectivity index (SI) 18 [3,4]) would be in summer. The diagram (Figure 1) clearly displays that in the summer months, especially in August, the target compound is present in greater quantities than in the other extracts. Furthermore, it can be deduced that *O*-tigloylcyclovirobuxeine-B (**1**) is only detectable in the variety with pointed leaves (Figure 1). Optimally, only leaves of *B*. *sempervirens* var. *arborescens* should be used for isolation in order to increase the yield.

In contrast, Cyclomicrophyllidine-B (**7**) occurs exclusively in the leaf extracts of *B. sempervirens* var. *suffruticosa* and exhibits a maximum content in May (Figure 2). This constituent displayed very potent antiplasmodial activity in a sub-micromolar range (IC_50_ 0.2 μM) with simultaneously high selectivity (SI 145) in our previous study [3], and hence represents a promising lead structure for the development of new antimalarial agents. It would therefore be required in larger quantities for subsequent investigations, e.g., in vivo studies. It can be concluded from the present data that the leaves of *B*. *sempervirens* var. *suffruticosa*, harvested in May, represent the optimal plant material for the isolation of **7**.

### 2.2. Differences between the Two Varieties and Seasonal Variability

To identify in a more holistic manner the major determinants of the chemical differences between the distinct harvesting times and the two varieties, a MVDA approach was chosen. Firstly, principal component analysis (PCA) was calculated for the whole bucket table. Appropriate scaling of the data is important for the validity of the resulting statistical models. A scaling method should be chosen to maximize the distance between the relevant variables (“buckets”), which will then also lead to a maximum discrimination between unrelated samples in the scores plot. PCA models were created from the bucket table using three different scaling methods (unit variance, Pareto and level scaling) or no scaling and compared with each other (Figure 3 and Appendix A).

In the present case, unit variance scaling (UVS; Appendix A) and Pareto scaling (PS; Figure 3) as well as the use of unscaled data (Appendix A) yielded models that discriminated in a similarly efficient manner between the samples of the two different *B. sempervirens* varieties. In all three cases, these samples were very efficiently distinguished on the first PC (PC1). Using unscaled data, intense signals (i.e., compounds occurring in large absolute concentrations) dominate the model very strongly, meaning that more subtle differences related to changes in constituents occurring at lower concentrations may be missed. On the other hand, in UVS, each bucket value is divided by the standard deviation of all values of the particular variable so that each bucket variable is assigned the same (=unit) variance. Thus, information on quantitative differences between the constituents is entirely lost [13]. Pareto scaling, where each bucket value is divided by the square root of the variable’s standard deviation, represents a useful compromise between these two extremes. This PCA model was therefore chosen for the analysis of further details.

Figure 3 shows the scores (**A**) and loadings (**B**) plot of the PCA model obtained with PS data. The second PC (PC2) is plotted vs. the first (PC1), which, taken together, explain roughly half (47.5%) of the total variance in the bucket table. It is very plain to see that PC1, explaining 28.9% of the total variance, distinguishes between the two varieties of *B. sempervirens*, which appear as very well separated groups (round dots with low scores on PC1 and triangles with high scores on that PC, for var. *suffruticosa* and *arborescens*, respectively). The second component (PC2, 18.6% variance explained) contains information on the temporal variability of the samples. The numerals in the scores plot represent months, and it is evident that samples taken during the summer months show high scores, while those of the winter months have low scores on PC2. As indicated by the circles in Figure 3A, the deviations between the two technical replicates taken in different measurement series can be considered as being small.

The PCA model clearly shows that the major sources of variance in the current dataset are (1) the different spectrum of constituents in the two varieties of *B*. *sempervirens* and (2) the gradual changes in constituents between the summer and winter months. Interestingly, these time-periodic differences become smaller in winter and show their maximum in the months of July and August.

#### 2.2.1. Identification of the Compounds Mainly Responsible for Differences in Varieties and Temporal Variation

The buckets (i.e., t_R_: *m/z* variables) mainly responsible for the variety- and time-dependent differences could be derived from the loadings plot of the PCA model with PS data (Figure 3B). Bucket variables showing great variance are localized in the periphery of the loadings plot, i.e., at extreme values for PC1 and PC2. Such variables are of particular interest in the evaluation, as they represent the main determinants of the differences between varieties and seasons. The analysis of the compounds represented by the highlighted buckets was then performed by inspecting the underlying +ESI-QqTOF MS/MS spectra, which were compared with literature data and authentic reference samples of isolated alkaloids from *B*. *sempervirens* [3]. It has been demonstrated previously that the ESI-MS spectra of *Buxus*-alkaloids contain diagnostic fragments related to the core skeletons’ substitution and degree of saturation. Such characteristic fragments [14,15,16] were compiled (Figure 4) and the MS^2^ spectra underlying the relevant buckets (Figure 5) were examined in this respect.

Seven buckets were located in the periphery of the loadings plot (Figure 5) and were thus shown to represent major characteristic varietal and seasonal constituents. The chromatographic and mass spectral data of the LC/MS analysis of the compounds are listed in Table 2.

Compounds **22** (11.74 min: 547.393 *m/z*) (Appendix A), **13** (9.39 min: 370.311 *m/z*) (Appendix A), **23** (11.81 min: 547.392 *m/z*) (Appendix A) and **12** (7.27 min: 400.323 *m/z*) (Appendix A) could be assigned to *N*-benzoyl-*O*-acetylbuxodienine-*E* (**22**), *E*-Cyclobuxophyllinine-M (**13**), *N*-benzoyl-*O*-acetylbuxadine-*E* (**23**) and N_b_-dimethylcycloxobuxoviricine (**12**), respectively, based on their exact molecular masses and fragmentation pattern. These four compounds have already been isolated in our prior work and their structure unambiguously established using NMR spectroscopy [3]. The identity of the compounds detected in the present study based on the PCA model was confirmed by identity of their retention time and mass spectra with these authentic reference compounds. In addition, compound **26** (10.27 min: 505.381 *m/z*) (Appendix A) could be assigned by comparison with literature data [17] and by analysis of its fragmentation pattern as *N*-benzoyl-cycloxo-buxin F (**26**), an alkaloid already known from *B*. s*empervirens*. The detailed fragmentation pathway is reported in Appendix A in the Appendix A.

Compounds **27** (12.53 min: 389.125 *m/z*) (Appendix A, Appendix A) and **28** (14.08 min: 414.361 *m/z*) (Appendix A) did not display the characteristic fragmentation of *Buxus*-alkaloids and could not be assigned.

Compounds **22**, **13** and **26** were highlighted by the PCA model (Figure 5) as characteristic constitutions of summer months with particularly high loadings on PC2, while compounds **13**, **26** and **23**, with high loadings on PC1, were indicated as characteristic compounds of *B*. *sempervirens* var. *arborescens*, distinguishing it from var. *suffruticosa*. In contrast, *N*_b_-dimethylcycloxobuxoviricin (**12**) turned out to be characteristic of *B*. *sempervirens* var. *suffruticosa* due to the far-left orientation in the loadings plot (Figure 5).

### 2.3. Distinction between the Leaves and Twigs

Having thus identified the main determinants of differences between the two varieties and of temporal variation in their pattern of constituents, it was an interesting question as to whether it would also be possible to identify constituents characteristic for a distinction between the leaves and twigs. In this case, the Pareto-scaled PCA model did not yield such clear results as described above for the distinction between varieties and time dependent changes. It can be seen in the scores plot of this model shown in Figure 3A that the twig samples (green) appear at the “lower ends” of the two variety groups, i.e., at more negative values of PC2. In that respect, they resemble leaf samples obtained in the winter months. A very similar behavior is also observed in the corresponding plot of PC2 vs. PC1 of the model obtained with unit variance-scaled (UVS) data (Appendix A). Quite interestingly, the distinction between leaf and twig samples becomes more evident when a scores plot of the third vs. second PC of the UVS model is inspected (see Figure 6A). In this plot, the samples of the two organs appear completely separate due to the differences from the leaf samples on PC2 and PC3. This is also the case in the second vs. first PC of the model with level scaling (Appendix A) but, interestingly, none of the PCs of the PS model represent this distinction in such a clear manner.

The corresponding loadings plot of the UVS model (Figure 6B) is more diffuse than that of the PS model (Figure 3B), i.e., it is less straightforward to identify variables with a particularly high impact on the distinction between sample groups. Therefore, a t-test was carried out in the software ProfileAnalysis 2.1 in order to identify characteristic buckets for the two different plant organs. Firstly, the signals of constituents that were only detectable in one sample type (leaf/twig) could be taken directly from the t-test result table as unique features (value count 0 for one of the two organ groups). Furthermore, a volcano plot was generated and analyzed to identify constituents that occur in both organs but differ significantly in quantity between these sample types.

#### 2.3.1. Identification of Constituents Occurring Exclusively in Leaves or Twigs (“Unique Features”)

For each group (leaf/twig), two unique features (Figure 7, compounds **29**–**32**) were analyzed, respectively, using the related chromatographic and mass spectral data (Table 3).

Compounds **29** (12.25 min: 489.386 *m/z*) and **30** (11.11 min: 535.392 *m/z*) occurred exclusively in the leaf extracts and could be considered as being characteristic of the leaf organ. The molecular formula of compound **29** was determined as C_33_H_48_N_2_O by LC/MS (Appendix A) and the core fragment *m/z* 323 indicated a C-3/C-20 diamine or amide with one additional substitution or double bond. The absence of the fragment *m/z* 297 illustrated that the double bond is not located between C-6 and 7 (Figure 4). It could be deduced from the characteristic fragmentation pattern in the +ESI-QqTOF MS/MS spectrum that compound **29** belongs to the 9β-19-cyclo-5α-pregnane series. Additionally, it could be derived from the MS analysis that only one of the two nitrogen substituents is basic (i.e., an amino group) ([M + 2H]^2+^ < [M + H]^+^), so that the other one should be part of an amide group. The fragments *m/z* 444 [M + H-(CH_3_)_2_NH]^+^ and 323 [444-C_6_H_5_CONH_2_]^+^ consistently indicated the neutral loss of a dimethylamino group and a benzamide moiety. To the best of our knowledge, no *Buxus*-alkaloid has been described so far with these features represented by the putative structure in Figure 7.

Another constituent characteristic only for leaf samples, compound **30** (Appendix A), on the grounds of its mass spectra, could be assigned to buxruguloid-B, a known constituent of *B*. *rugulosa* [18]. The fragmentation pathway (Appendix A) and the occurrence in *B*. *sempervirens* are communicated for the first time in this work.

For the twig extracts, compounds **31** (12.11 min: 577.401 *m/z*) and **32** (11.94 min: 366.280 *m/z*) were highlighted as unique features. Compound **31** displayed a characteristic fragmentation pattern (Appendix A) very similar to compounds **22** and **23**. From the UV spectrum (λ_max_ 224 nm; Appendix A), it could be deduced that compound **30** possesses an isolated double bond system as the structure of compound **23** [17] and not a conjugated diene system as compound **22** (λ_max_ 237, 245 and 253 nm) [3]. The molecular formula of compound **30** was determined as C_36_H_52_N_2_O_4_ by LC/MS (Appendix A). The neutral loss of 151 Da (*m/z* 472→321) indicated the presence of an additional methoxy group substitution at the secondary benzamide structure [19], in contrast to *N*-benzoyl-*O*-acetylbuxadine-*E* (**23**) (Figure 7). The detailed fragmentation pathway is reported in Appendix A in the Appendix A. To the best of our knowledge, a compound with this structure has not been described in the literature up to present.

Compound **32** possesses the molecular formula C_25_H_35_NO and nine double bond equivalents according to LC/MS analysis (Appendix A). The first fragment *m/z* 335 [366-CH_3_NH_2_]^+^ in the +ESI-QqTOF MS/MS spectrum (Appendix A) indicated the neutral loss of a monomethylamino group (−31 Da). The subsequent fragmentation pathway could not be elucidated within this work. According to these mass spectral data, we suggest that compound **32** could be assigned to the known spiro-cyclopentane structure Spirofornabuxin [20], the only constituent of *B*. *sempervirens* with this elemental composition described so far.

#### 2.3.2. Identification of Compounds with Different Content in Leaves and Twigs

The result of the t-test calculation is illustrated in the volcano plot (Figure 8), which displays differences between samples obtained from twigs and leaves. In the volcano plot, the negative decadic logarithm of the *p*-value (significance) is plotted against the binary logarithm of the fold change (quantity change of a compound between both groups). Of particular interest are signals (buckets) with a high fold change and a significant *p*-value, although greater attention was paid to the fold change. This eliminates signals that show significant differences between sample types, which are, however, irrelevant due to their small intensity. Once more, the LC/MS spectra (Table 4) of the compounds underlying the interesting buckets were analyzed.

*E*-cyclobuxophyllinine-M (**13**) and *N*-benzoyl-*O*-acetylbuxadine-*E* (**23**) have already been identified as characteristic of *B*. *sempervirens* var. *arborescens* (Section 2.2.1), and could additionally be detected in significantly higher concentrations in the leaves compared to the twigs in the volcano plot (Figure 8).

Compounds **33** (11.92 min: 533.376 *m/z*) (Appendix A), **34** (12.14 min: 533.377 *m/z*) (Appendix A) and **19** (5.28 min: 346.313 *m/z*) (Appendix A) could be assigned to buxusemine-H (**33**) [21], buxusemine-L (**34**) [21] and irehine (**19**) [22] by means of the analysis of the chromatographic and mass spectral data. Irehine has already been isolated by us in a previous study [3], so that the assignment of compound **19** could be confirmed by direct comparison with an authentic sample.

For compound **35** (12.21 min: 491.401 *m/z*), the molecular formula was determined as C_33_H_50_N_2_O by LC/MS (Appendix A). Due to the core fragment *m/z* 325, compound **35** could be assigned to the fully saturated C-3/C-20 diamines/amides (Figure 4). The fragments *m/z* 446 [M + H-(CH_3_)_2_NH]^+^ and 325 [446-C_6_H_5_CONH_2_]^+^ indicated a substitution with a dimethylamino group and a secondary benzamide. The complete fragmentation pathway is shown in Appendix A in the Appendix A. The location of the amino and amide groups at C-3 and C-20 could not be assigned on the grounds of the mass spectra alone (Figure 8). Both possible structures were, to the best of our knowledge, not reported previously.

In contrast, compounds **36** (11.22 min: 384.327 *m/z*), **37** (12.81 min: 285.162 *m/z*) and **38** (12.54 min: 384.259 *m/z*) were present in the twigs in significantly higher quantities than in the leaves studied (Figure 9). The chromatographic and mass spectral data of these three constituents of *B*. *sempervirens* are listed in Table 5.

Compound **36** possesses the molecular formula C_26_H_41_NO according to LC/MS analysis (Appendix A). The fragment at *m/z* 339 [M-((CH_3_)_2_NH)]^+^ resulted from the neutral loss of a dimethylamino group (−45 Da), while the subsequent fragment at *m/z* 321 [339-H_2_O]^+^ indicated the presence of a hydroxyl group. The core fragment *m/z* 295 in the related +ESI-QqTOF MS/MS spectrum indicated a *Buxus*-alkaloid with a double bond between C-6 and C-7 (Figure 4). The derived structure (Figure 9) could not be found in the literature, meaning that we assume, to the best of our knowledge, that compound **36** has not been described before.

Compounds **37** (Appendix A) and **38** (Appendix A) did not show the characteristic fragmentation of *Buxus*-alkaloids and hence could not be assigned based on literature data.

## 3. Materials and Methods

### 3.1. Extraction of Plant Material

The plant material, as in our previous studies [3,4], was obtained from a hedge growing on a private estate in Havixbeck, Germany (GPS coordinates are available from the corresponding author on request). Within one year (February 2019–January 2020), shoot tips of the same box hedge as in our previous studies [3,4,14] were harvested each month in order to create a seasonal profile of *B*. *sempervirens* L. This hedge is composed of two varieties with rounded and pointed leaves, respectively (Figure 10), with the one with pointed leaves being clearly predominant in proportion. Based on the literature [5,6], the variety with pointed leaves was identified as *B*. *sempervirens* var. *arborescens* L., and the variety with rounded leaves was identified as *B*. *sempervirens* var. *suffruticosa* L. In addition to the various harvesting times and the two different varieties, a distinction was made between the organs by separating leaves from twigs.

The plant material was air-dried at room temperature and ground with an IKA MF 10 basic mill (Staufen, Germany) to 1 mm mesh size. Of each sample, 3.5 g of ground drug material was extracted with 35 mL of dichloromethane (DCM) on a magnetic stirrer (RCT classic, IKA, Staufen, Germany) for 30 min at room temperature. The extraction was repeated twice with 35 mL of the same solvent and the entire extract was evaporated to dryness on the rotary evaporator (R-210, Büchi, Flawil, Switzerland).

A total of 30 extracts (12 leaf extracts from each variety for each month and 3 twig extracts from each variety for the months of February, September and December) were obtained and analyzed by LC/MS (Ultimate RS 3000, Dionex, Sunnyvale, CA, USA).

### 3.2. UHPLC/+ESI-QqTOF-Mass Spectrometry

Each extract was dissolved in methanol (MeOH) at a concentration of 10 mg/mL for recording a fingerprint chromatogram (full scan MS) by UHPLC/+ESI-QqTOF-MS (Ultimate RS 3000, Dionex, Sunnyvale, CA, USA) using the same method and parameters as previously described [4]. As an internal standard (IS), 10 µL of papaverine-HCL solution (0.25 mg/mL) was added to reveal deviations in the injection and to detect a possible loss of sensitivity over the measurement. The injection volume was 1 µL. In order to obtain more precise information regarding the fragmentation behavior, MS2 experiments were also recorded from some samples in advance. These runs also served to saturate the stationary phase and thus achieve retention time constancy. In these cases, 2 µL were injected. The extracts were analyzed in a continuous sequence for better comparability. After the matrix saturation runs, a series of measurements was carried out seamlessly in chronological order from a seasonal point of view (from January to December), while the sample sequence was chosen randomly for the second series of measurements in order to be able to detect possible carry-over. The sample BS01_20_B_s (January 2020, *B*. *sempervirens* var. *arborescens*, leaves) was chosen as a quality control (QC) and analyzed after every 20 sample injections. Likewise, a blank run (MeOH) was carried out after 20 injections but in alternating order with the QC so that either QC or Blank were analyzed after every 10th sample injection.

### 3.3. Pretreatment of LC/MS Data

The pretreatment of LC/MS data and the creation of a descriptor table (“bucket table”) were carried out with the software ProfileAnalysis 2.1 (Bruker Daltonik GmbH, Bremen, Germany) in the same way as in our previous study [14]. A value count of group (organ: leaves vs. twigs) attributes within bucket ≥ 10% was selected. The intensity of the resulting buckets was normalized to the internal standard. The bucket table thus obtained was used to calculate the PCA models.

### 3.4. PCA Modeling

The data of the bucket table (1200 bucket variables) were mean-centered by default in ProfileAnalysis before the models were created. In addition, three different scaling methods (unit variance, Pareto and level scaling) and the use of the unscaled data (none) for model calculation were tested (Section 2.2 and 2.3). A confidence level of 95% was applied in each case and the data matrix was cross-validated (leave-one-out cross validation) for the calculation.

### 3.5. Volcano Plot

Based on the bucket table, a t-test was carried out in the ProfileAnalysis software (version 2.1, Bruker Daltonik GmbH, Bremen, Germany) in order to find constituents distinguishing leaves from twigs. The fold change, which indicates the concentration change of a substance or a metabolite in different samples, and the value count, which points out in how many samples the bucket are detectable, were determined.

The unique features (value count of 0 for twig or leaf) were taken directly from the t-test result table, while the compounds that showed significant differences in content between the twig and leaf extracts were highlighted in the volcano plot. Buckets with a high fold change (|log_2_(fold change)| ≥ 2.32 for leaf and |log_2_(fold change)| ≥ 2 for twig) and a low *p*-value (<0.05; *p*-value ≤ 0.05 (−log_10_(*p*) ≥ 1.3)) [23] were considered in the evaluation.

## 4. Conclusions

The present study, for the first time, shows the main chemical differences between the investigated varieties, *B. sempervirens* var. *arborescens* and *suffruticosa*. The results will allow the selection of the right plant variety and optimal harvesting time to obtain certain alkaloids that are, e.g., of interest because of a particular biological activity. The results of the MVDA of the complex dataset by principal component and t-test/volcano plot analysis yield a holistic picture of differences in the main constituents of the two varieties and their leaves and twigs. In addition to this valuable information, these detailed comparative studies also provide interesting data on the seasonal changes in secondary metabolites in the plants studied. Thus, marker substances for the different varieties such as *N*-benzoyl-*O*-acetylbuxadine-*E* (**23**) for *B*. *sempervirens* var. *arborescens* L. and *N*_b_-dimethylcycloxobuxoviricine (**12**) for *B*. *sempervirens* var. *suffruticosa* L. could be identified. Similarly, leaves and twigs can be distinguished by the occurrence of buxruguloid-B (**30**) and compound **31** (new alkaloid), respectively. It is to be noted that, based on detailed LC/MS analysis, +ESI fragmentation pathways of various known and several putative new *Buxus*-alkaloids were established here for the first time, which will enable the rapid dereplication of these natural products in future studies.

## Figures and Tables

**Figure 1 molecules-27-00082-f001:**
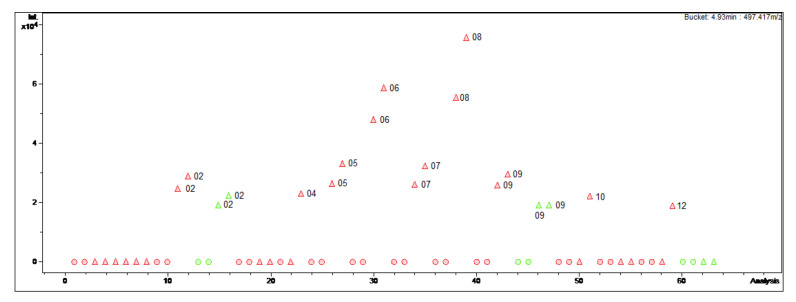
Bucket Statistic of *O*-tigloylcyclovirobuxeine-B ((**1**); 4.93 min: 497.417 *m/z*) in ProfileAnalysis 2.1. Legend: round dots = *B*. *sempervirens* var. *suffruticosa*; triangle = *B*. *sempervirens* var. *arborescens*; red = leaf; green = twig; number = month.

**Figure 2 molecules-27-00082-f002:**
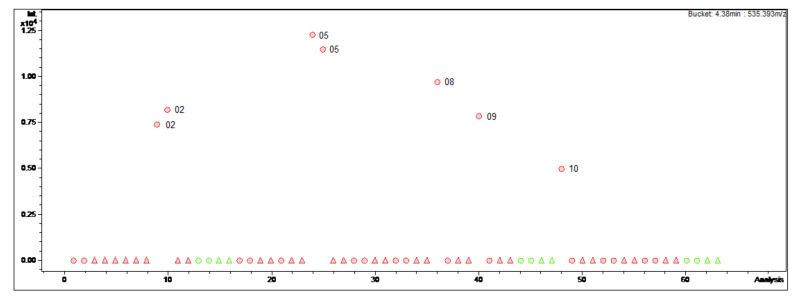
Bucket statistic of Cyclomicrophyllidine-B ((**7**); 4.38 min: 535.393 *m/z*) in ProfileAnalysis 2.1. Legend: round dots = *B*. *sempervirens* var. *suffruticosa*; triangle = *B*. *sempervirens* var. *arborescens*; red = leaf; green = twig; number = month.

**Figure 3 molecules-27-00082-f003:**
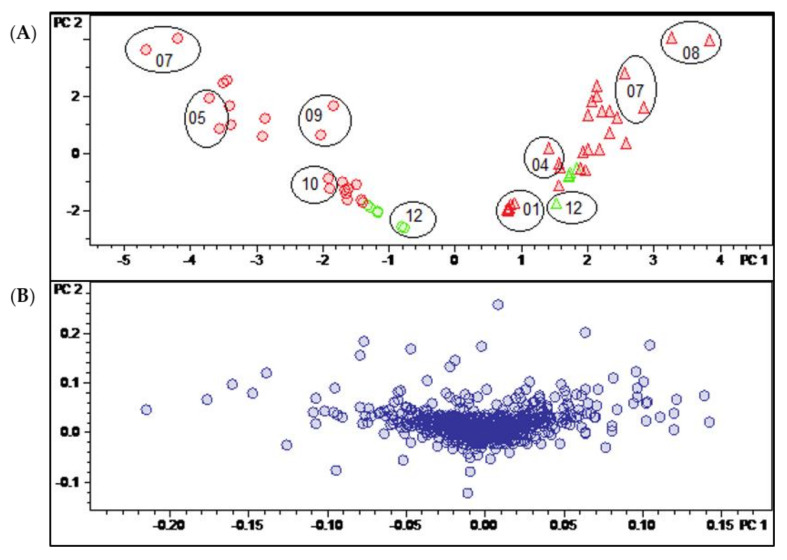
(**A**) Scores and (**B**) loadings plot of the PCA model obtained with Pareto-scaled variable data. PC1 (explaining 28.9% of the total variance in the data) and PC2 (explaining further 18.6% of the total variance). Legend for (**A**): round dots = *B*. *sempervirens* var. *suffruticosa*; triangle = *B*. *sempervirens* var. *arborescens*; red = leaf; green = twig; number = month; circle = related technical replicates (*n* = 2; quality control (QC) *n* = 4).

**Figure 4 molecules-27-00082-f004:**
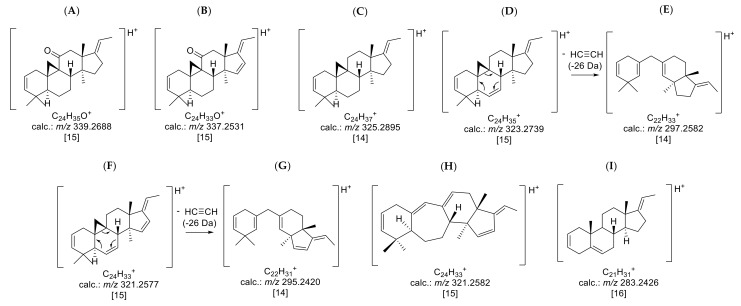
Prominent core fragments of the investigated *nor*-triterpene alkaloids of the *nor*-cycloartane type from *B*. *sempervirens*: (**A**) core fragment *m/z* 339 for fully saturated C-3/C-20 diamines with a ketone substituent; (**B**) core fragment *m/z* 337 for C-3/C-20 diamines with a ketone substituent possessing one additional substitution or double bond; (**C**) core fragment *m/z* 325 for fully saturated C-3/C-20 diamines; (**D**) core fragment *m/z* 323 for C-3/C-20 diamines possessing one additional substitution or double bond; (**E**) core fragment *m/z* 297 for C-3/C-20 diamines with a Δ^6^; (**F**) core fragment *m/z* 321 for C-3/C-20 diamines doubly substituted or unsaturated; (**G**) core fragment *m/z* 295 for C-3/C-20 diamines with a Δ^6^ and one further substitution or unsaturation; (**H**) core fragment *m/z* 321 for C-3/C-20 diamines of (9-(10→19))abeo-5α-pregnane possessing one additional substitution or double bond; (**I**) core fragment *m/z* 283 for steroidal alkaloids with a C-3/C-20 substitution possessing one additional double bond or substitution. The postulated positions of unsaturation and substitution are based on the already known structures of compounds **19**, **22** and **26 [3,17]**.

**Figure 5 molecules-27-00082-f005:**
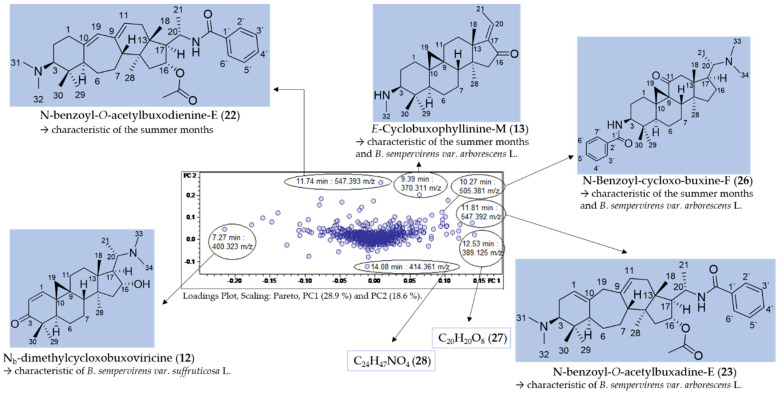
Assignment of buckets showing great variance in the loadings plot of PC2 vs. PC1 (compared with Figure 3B).

**Figure 6 molecules-27-00082-f006:**
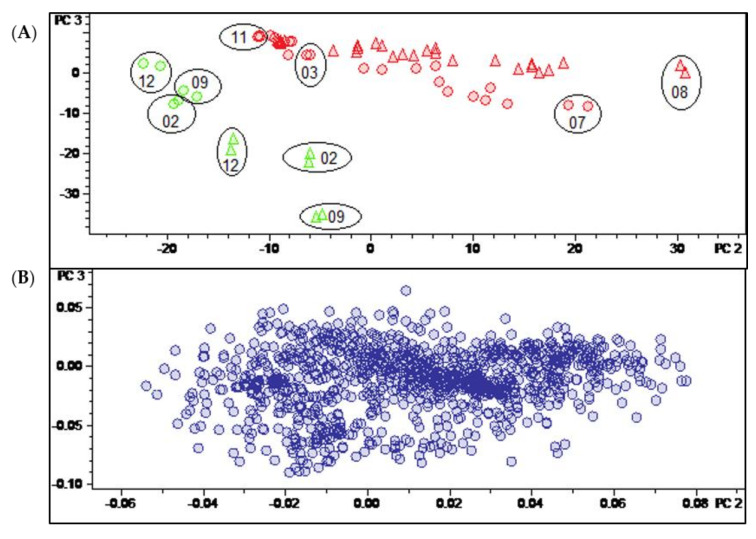
(**A**) Scores and (**B**) loadings plot of the PCA model obtained with unit variance-scaled variable data. PC2 (explaining 12.6% of the total variance in the data) and PC3 (explaining further 7.9% of the total variance). Legend for (**A**): round dots = *B*. *sempervirens* var. *suffruticosa*; triangle = *B*. *sempervirens* var. *arborescens*; red = leaf; green = twig; number = month; circle = related technical replicates (*n* = 2; quality control (QC) *n* = 4).

**Figure 7 molecules-27-00082-f007:**
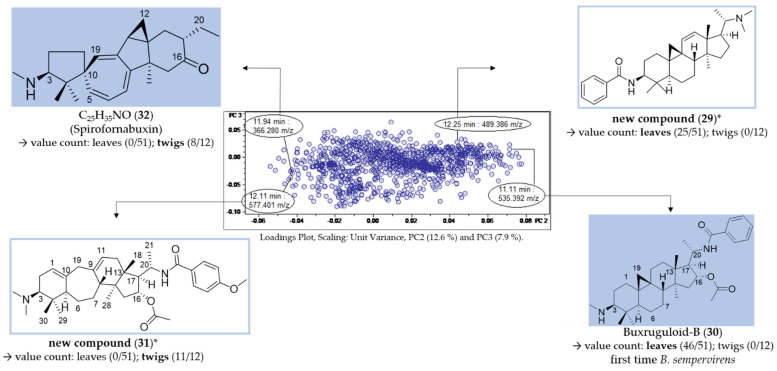
Assignment of unique features in the loadings plot of PC3 vs. PC2 (compare Figure 6B). * Structural assignment based only on mass spectral data; stereochemistry tentatively assumed in analogy with known compounds [3].

**Figure 8 molecules-27-00082-f008:**
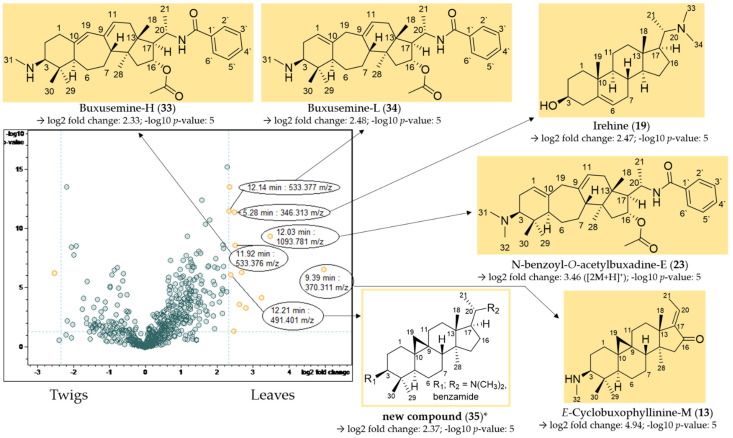
Volcano plot of the comparison of the twig with the leaf extracts. Buckets with a high fold change (|log_2_(fold change)| ≥ 2.32) and a low *p*-value ≤ 0.05 (−log_10_(*p*) ≥ 1.3) are shown in yellow. Assignment of compounds that are present in significantly higher concentrations in the leaf extracts compared to the twig extracts. * Structural assignment based only on mass spectral data; stereochemistry tentatively assumed in analogy with known compounds [3].

**Figure 9 molecules-27-00082-f009:**
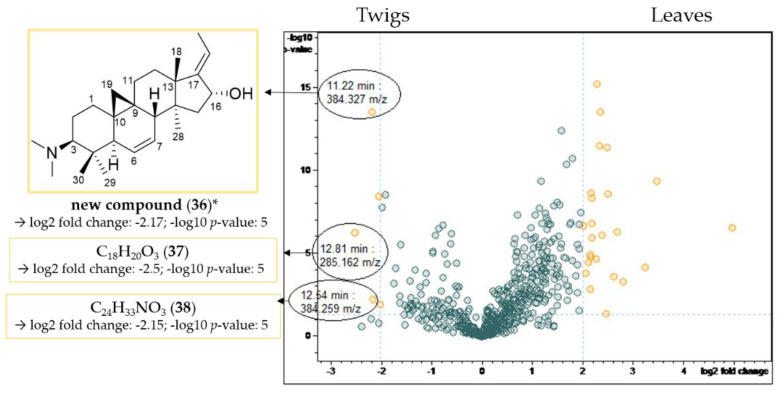
Volcano plot of the comparison of the twig with the leaf extracts. Buckets with a high fold change |log_2_(fold change)| ≥ 2 and a low *p*-value ≤ 0.05 (−log_10_(*p*) ≥ 1.3) are shown in yellow. Assignment of compounds that are present in higher concentrations in the twig extracts compared to the leaf extracts. * Structural assignment based only on mass spectral data; stereochemistry tentatively assumed in analogy with known compounds [3].

**Figure 10 molecules-27-00082-f010:**
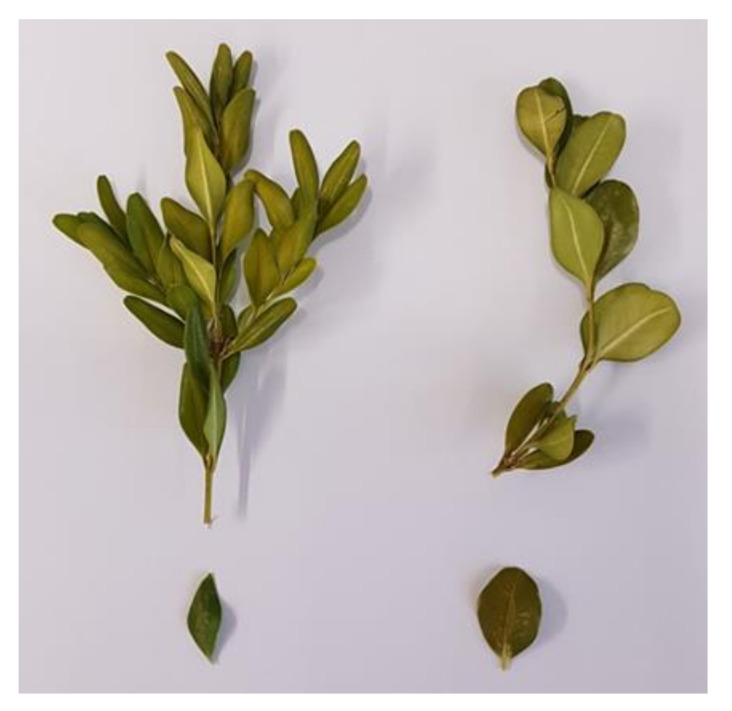
Left: Shoot tip and single leaf of *B*. *sempervirens* var. *arborescens* L.; right: shoot tip and single leaf of *B*. *sempervirens* var. *suffruticosa* L.

**Table 1 molecules-27-00082-t001:** Assignment of the origin of the previously isolated *Buxus*-alkaloids (**1–25**) [3].

Cmpd.	*Buxus*-Alkaloid	Bucket	Variety	Month of Maximal Content	Organ
**1**	*O*-tigloylcyclovirobuxeine-B	4.93 min: 497.417 *m/z*	*arborescens*	August	leaf > twig
**2**	Cyclovirobuxeine-B	3.64 min *	*arborescens* > *suffruticosa*	August	leaf > twig
**3**	*O*-tigloylcyclomicrophylline-B	4.46 min *	*arborescens*	August	leaf > twig
**4**	*O*-tigloylcyclomicrophylline-A	4.82 min: 527.392 *m/z*	*arborescens*	August	leaf
**5**	Cyclomicrophylline-A	3.07 min: 445.390 *m/z*	*suffruticosa*	July	leaf
**6**	Cyclomicrophyllidine-A	4.64 min: 549.410 *m/z*	*suffruticosa*	July	leaf
**7**	Cyclomicrophyllidine-B	4.38 min: 535.393 *m/z*	*suffruticosa*	May	leaf
**8**	*O*-benzoyl-cycloprotobuxoline-d	5.28 min: 507.401 *m/z*	*arborescens* > *suffruticosa*	August	leaf > twig
**9**	*N*-benzoyl-*O*-acetyl-cycloxo-buxoline-F	7.16 min: 563.387 *m/z*	*arborescens* > *suffruticosa*	August	leaf > twig
**10**	*N*-benzoyl-cycloxo-buxoline-F	7.15 min: 521.379 *m/z*	*arborescens*	August	leaf > twig
**11**	29-hydroxy-cyclomikuranine-l	5.00 min: 418.334 *m/z*	*suffruticosa* > *arborescens*	July	leaf > twig
**12**	*N*_b_-dimethylcycloxo-buxoviricine	7.27 min: 400.323 *m/z*	*suffruticosa* > *arborescens*	July	leaf > twig
**13** + **14**	(*E*)-cyclobuxophyllinine-M + (*Z*)-cyclobuxophyllinine-M	9.39 min: 370.312 *m/z*	*arborescens* > *suffruticosa*	August	leaf
**15** + **16**	(*E*)-cyclosuffrobuxinine-M + (*Z*)-cyclosuffrobuxinine-M	7.88 min: 354.281 *m/z*	*arborescens* ≈ *suffruticosa*	July	leaf ≈ twig
**17**	Cyclomicrobuxinine	5.05 min: 372.292 *m/z*	*arborescens* > *suffruticosa*	August	leaf ≈ twig
**18**	Cyclomicrobuxine	5.19 min: 386.307 *m/z*	*arborescens* ≈ *suffruticosa*	September	leaf ≈ twig
**19**	Irehine	5.28 min: 346.313 *m/z*	*suffruticosa* > *arborescens*	July	leaf > twig
**20**	16-α-hydroxybuxaminone	4.91 min: 400.324 *m/z*	*arborescens* > *suffruticosa*	September	leaf ≈ twig
**21**	*N*_20_-acetylbuxamine-*E*	7.13 min: 427.370 *m/z*	*arborescens* > *suffruticosa*	August	leaf > twig
**22**	*N*-benzoyl-*O*-acetylbuxodienine-*E*	11.74 min: 547.393 *m/z*	*arborescens* ≈ *suffruticosa*	July	leaf > twig
**23**	*N*-benzoyl-*O*-acetylbuxadine-*E*	11.81 min: 547.392 *m/z*	*arborescens*	August	leaf > twig
**24**	*N*_20_-acetylbuxadine-G	9.83 min: 413.354 *m/z*	*arborescens* > *suffruticosa*	August	leaf
**25**	17,20-dihydroxybuxadine-M	6.56 min *	*arborescens*	August	leaf > twig

Cmpd.: compound; * Compounds **2**, **3** and **25** were not represented as buckets in the bucket table, so the occurrence of these alkaloids in the different extracts was evaluated via extracted ion chromatograms (EICs) for the respective [2M + 2H]^2+^ ions (*m/z* 208.1902 (**2**), 257.2174 (**3**)) and [M + H]^+^ ion (*m/z* 388.3272 (**25**)).

**Table 2 molecules-27-00082-t002:** LC/MS-characteristics of compounds with great variance (compare Figure 5).

Cmpd.	Bucket	Adduct Ions	Structural Formula(DBE)	Core Fragment(s) *m/z*	Identified *Buxus*-Alkaloid
**22**	11.74 min: 547.393 *m/z*	[M + 2H]^2+^ < [M + H]^+^	C_35_H_50_N_2_O_3_ (12)	321	*N*-benzoyl-*O*-acetylbuxodienine-*E*
**13**	9.39 min: 370.311 *m/z*	[M + H]^+^	C_25_H_39_NO (7)	339	*E*-Cyclobuxophyllinine-M
**26**	10.27 min: 505.381 *m/z*	[M + 2H]^2+^ < [M + H]^+^	C_33_H_48_N_2_O_2_ (11)	339	*N*-benzoyl-cycloxo-buxine-F
**23**	11.81 min: 547.392 *m/z*	[2M + H]^+^ < [M + 2H]^2+^ < [M + H]^+^	C_35_H_50_N_2_O_3_ (12)	321	*N*-benzoyl-*O*-acetylbuxadine-*E*
**27**	12.53 min: 389.125 *m/z*	[M + H]^+^	C_20_H_20_O_8_ (11)	n.i.	n.i.
**28**	14.08 min: 414.361 *m/z*	[M + H]^+^	C_24_H_47_NO_4_ (2)	n.i.	n.i.
**12**	7.27 min: 400.323 *m/z*	[M + H]^+^	C_26_H_41_NO_2_ (7)	337	*N*_b_-dimethylcycloxobuxoviricine

Cmpd.: compound; DBE: double bond equivalent; n.i.: not identified.

**Table 3 molecules-27-00082-t003:** LC/MS-characteristics of unique features (compare Figure 7).

Cmpd.	Bucket	Adduct Ions	Structural Formula (DBE)	Core Fragment(s) *m/z*	Identified *Buxus*-Alkaloid
**29**	12.25 min: 489.386 *m/z*	[M + 2H]^2+^ < [M + H]^+^	C_33_H_48_N_2_O (11)	323	new compound
**30**	11.11 min: 535.392 *m/z*	[M + 2H]^2+^ < [M + H]^+^	C_34_H_50_N_2_O_3_ (11)	323	Buxruguloid-B
**31**	12.11 min: 577.401 *m/z*	[M + 2H]^2+^ < [M + H]^+^	C_36_H_52_N_2_O_4_ (12)	321	new compound
**32**	11.94 min: 366.280 *m/z*	[M + H]^+^	C_25_H_35_NO (9)	n.i.	(Spirofornabuxin)

Cmpd.: compound; DBE: double bond equivalent; n.i.: not identified.

**Table 4 molecules-27-00082-t004:** LC/MS-characteristics of compounds displaying a high fold change and a low *p*-value (compare Figure 8).

Cmpd.	Bucket	Adduct Ions	Structural Formula (DBE)	Core Fragment(s) *m/z*	Identified *Buxus*-Alkaloid
**33**	11.92 min: 533.376 *m/z*	[M + 2H]^2+^ < [M + H]^+^	C_34_H_48_N_2_O_3_ (12)	321	Buxusemine-H
**34**	12.14 min: 533.377 *m/z*	[M + 2H]^2+^ < [M + H]^+^	C_34_H_48_N_2_O_3_ (12)	321	Buxusemine-L
**19**	5.28 min: 346.313 *m/z*	[M + H]^+^	C_23_H_39_NO (5)	283	Irehine
**23**	12.03 min: 1093.781 *m/z*	[2M + H]^+^ < [M + 2H]^2+^ < [M + H]^+^	C_35_H_50_N_2_O_3_ (12)	321	*N*-benzoyl-*O*-acetylbuxadine-*E*
**13**	9.39 min: 370.311 *m/z*	[M + H]^+^	C_25_H_39_NO (7)	339	*E*-Cyclobuxophyllinine-M
**35**	12.21 min: 491.401 *m/z*	[M + 2H]^2+^ < [M + H]^+^	C_33_H_50_N_2_O (10)	325	new compound

Cmpd.: compound; DBE: double bond equivalent; n.i.: not identified.

**Table 5 molecules-27-00082-t005:** LC/MS-characteristics of compounds displaying a high fold change and a low *p*-value (compare Figure 9).

Cmpd.	Bucket	Adduct Ions	Structural Formula (DBE)	Core Fragment(s) *m/z*	Identified *Buxus*-Alkaloid
**36**	11.22 min: 384.327 *m/z*	[M + H]^+^	C_26_H_41_NO (7)	321; 295	new compound
**37**	12.81 min: 285.162 *m/z*	[M + Na]^+^ < [M + H]^+^	C_18_H_20_O_3_ (9)	n.i.	n.i.
**38**	12.54 min: 384.259 *m/z*	[2M + Na]^+^ < [M + Na]^+^ < [M + H]^+^	C_24_H_33_NO_3_ (9)	n.i.	n.i.

Cmpd.: compound; DBE: double bond equivalent; n.i.: not identified.

## Data Availability

The data presented in this study are contained within the article and Appendix A. Raw data of the LC/MS analyses and the bucket table derived from these data are available from the corresponding author on request.

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
