# Peer review of "Investigation of the Variability of Alkaloids in Buxus sempervirens L. Using Multivariate Data Analysis of LC/MS Profiles"

_molecules, 2021, doi:10.3390/molecules27010082_

Round 1
Reviewer 1 Report
In the manuscript authored by Lara U. Szabó et al. authors performed a very interesting descriptive study highlighting alkaloids profile seasonal variation of two different varieties of Buxus sempervirens L. (arborescens and suffruticosa).
Authors identified the optimal harvesting condition to optimize the isolation of particular Buxus-alkaloids of interest.
The manuscript is well-designed and structured, despite herebelow I reported my comments and suggestions:
-I suggest authors to include in method section the geographical localization of harvesting place and also soil condition if available.
-figures quality need to be improved. sometimes it is hard to distinguish triangel from dots expecially in figure 2.
-I suggest authors to include a summary table reporting the main seasonal differences observed between the different varieties of Buxus sempervirens L. according to the alkaloids identified. As for example, when it is the optimal harvesting time for the isolation of N- benzyl-N-acetyl buxadine-E for B. sempervirens?
-Authors should further discuss how this new awearness in seasonal alkaloid profile variation could improve the biological or medical application of Buxus sempervirens L. .
Author Response
Reviewer 1:
In the manuscript authored by Lara U. Szabó et al. authors performed a very interesting descriptive study highlighting alkaloids profile seasonal variation of two different varieties of Buxus sempervirens L. (arborescens and suffruticosa).
Authors identified the optimal harvesting condition to optimize the isolation of particular Buxus-alkaloids of interest.
The manuscript is well-designed and structured, despite herebelow I reported my comments and suggestions:
-I suggest authors to include in method section the geographical localization of harvesting place and also soil condition if available.
The location was added. Since this is on the private estate of the corresponding author, we do not wish to publish exact GPS coordinates, but they can be obtained from the corresponding author on request which was noted in the manuscript.
No data on soil constitution exist. We are sorry about this.
-figures quality need to be improved. sometimes it is hard to distinguish triangel from dots expecially in figure 2.
We are extremely sorry but the graphics quality depends entirely on the software, which does not allow to increase the symbol size. However, since this is an online journal, it is quite easy to zoom in on the diagrams so that the triangles and circles are very easy to distinguish. We apologize that we cannot improve the Figures’ quality for technical reasons. [Note to Editorial Office: If the reviewer insists on this point, Figures 1 and 2 might be expanded to the full page width. However, we found that this looks quite awkward because there would be a lot of “free space” on the figures in that case].
-I suggest authors to include a summary table reporting the main seasonal differences observed between the different varieties of Buxus sempervirens L. according to the alkaloids identified. As for example, when it is the optimal harvesting time for the isolation of N- benzyl-N-acetyl buxadine-E for B. sempervirens?
The reviewer may have overlooked that exactly such information is already available in Table 1. For example, N-benzyl-N-acetyl buxadine-E would optimally be obtained from leaves of var. arborescens in the month of August. This information is reported for all compounds in table 1, which are the main alkaloids of the plants under study.
-Authors should further discuss how this new awearness in seasonal alkaloid profile variation could improve the biological or medical application of Buxus sempervirens L. .
We are not quite sure what the reviewer means here. We have discussed the impact of the presented results in the conclusions section. We do not see what else could be said here without potentially over-interpreting and hypothesizing too much or being overly repetitive. So we would rather leave the discussion unchanged, unless the reviewer can give us a more detailed hint what is meant here.
We thank the reviewer for the very positive overall assessment and the helpful suggestions for improvement.
Reviewer 2 Report
I suggested performing additional statistic analyses such as PLS-DA to complete the discussion. Also, please check some typos and grammatical errors. Finally, I recommend the publication of this manuscript in Molecules due to its high quality and promissory results.
Author Response
Reviewer 2:
I suggested performing additional statistic analyses such as PLS-DA to complete the discussion. Also, please check some typos and grammatical errors. Finally, I recommend the publication of this manuscript in Molecules due to its high quality and promissory results.
We believe that we have performed the right statistical analyses to serve our purpose as described in the manuscript. Certainly, using different statistical methods such as supervised learning (e.g. PLS or PLS-DA) can be helpful to answer certain questions. However, we hope that the reviewer will agree that adding results of other methods to this overall consistent and conclusive work would make the manuscript more complicated and might distract from the actual results we wish to report. However, we thank the reviewer for the hint towards PLS-DA which we will certainly try in case we need to extract further details from this data set.
A few typos were also found and corrected (all changes are tracked in the manuscript).
We thank the reviewer for the very positive overall assessment and the thoughtful suggestion for improvement.
Reviewer 3 Report
The manuscript titled ”Investigation of the Variability of Alkaloids in Buxus sempervirens L. using Multivariate Data Analysis of LC/MS Profiles” concerns the influence of vegetation season and variety of B. sempervirens L. The introduction gives the justification for undertaking this study, methods are described in proper detail, the results are presented and discussed logically. The results of this study could be of interest for researches looking for optimization of acquiring alkaloids with potential therapeutic activity.
Author Response
Reviewer 3:
The manuscript titled ”Investigation of the Variability of Alkaloids in Buxus sempervirens L. using Multivariate Data Analysis of LC/MS Profiles” concerns the influence of vegetation season and variety of B. sempervirens L. The introduction gives the justification for undertaking this study, methods are described in proper detail, the results are presented and discussed logically. The results of this study could be of interest for researches looking for optimization of acquiring alkaloids with potential therapeutic activity.
We thank the reviewer for the very positive overall assessment.
Round 2
Reviewer 1 Report
Authors adequately reply to most of the requests reported from the reviewer. The manuscript has been slightly modified and now can be considered for a possible publication.